# Breastfeeding initiation and duration through the COVID-19 pandemic, a linked population-level routine data study: the Born in Wales Cohort 2018–2021

Hope Eleri Jones [ID] , Mike J Seaborne, Mohamed R Mhereeg, Michaela James, Natasha L Kennedy, Amrita Bandyopadhyay, Sinead Brophy

National Centre for Population Health and Wellbeing Research, Faculty of Medicine, Health & Life Science, Swansea University Medical School, Swansea, UK

**Correspondence to**
Hope Eleri Jones; h.e.jones@swansea.ac.uk

## ABSTRACT

**Objectives** The WHO recommends exclusive breastfeeding for the first 6 months of life. This study aimed to examine the impact the pandemic had on breastfeeding uptake and duration, and whether intention to breastfeed is associated with longer duration of exclusive breastfeeding.

**Methods** A cohort study using routinely collected, linked healthcare data from the Secure Anonymised Information Linkage databank. All women who gave birth in Wales between 2018 and 2021 recorded in the Maternal Indicators dataset were asked about intention to breastfeed. These data were linked with the National Community Child Health Births and Breastfeeding dataset to examine breastfeeding rates.

**Results** Intention to breastfeed was associated with being 27.6 times more likely to continue to exclusively breastfeed for 6 months compared with those who did not intend to breastfeed (OR 27.6, 95% CI 24.9 to 30.7). Breastfeeding rates at 6 months were 16.6% prepandemic and 20.5% in 2020. When compared with a survey population, the initial intention to breastfeed/not breastfeed only changes for about 10% of women.

**Conclusion** Women were more likely to exclusively breastfeed for 6 months during the pandemic compared with before or after the pandemic. Arguably, interventions which enable families to spend more time with their baby such as maternal and paternal leave may help improve breastfeeding duration. The biggest predictor of breastfeeding at 6 months was intention to breastfeed. Therefore, targeted interventions during pregnancy to encourage motivation to breastfeed could improve duration of breastfeeding.

## INTRODUCTION

The health and social benefits of breastfeeding are already well established.[1] Breastfeeding provides short-term and long-term physical, economic and environmental improvements for children, women and society. Benefits to breastfed babies include fewer infections, increased intelligence,

> **WHAT IS ALREADY KNOWN ON THIS TOPIC**
> ⇒ The current WHO recommendations indicate that mothers should provide exclusive breastfeeding to their babies during the first 6 months of life. However, the UK has one of the lowest breastfeeding rates in the world.
>
> **WHAT THIS STUDY ADDS**
> ⇒ Breastfeeding rates at 6 months were significantly higher during the COVID-19 pandemic. Black mothers were significantly more likely to exclusively breastfeed for 6 months. Intention to breastfeed was the strongest predictor of exclusive breastfeeding at 6 months.
>
> **HOW THIS STUDY MIGHT AFFECT RESEARCH, PRACTICE OR POLICY**
> ⇒ Targeted interventions during pregnancy to encourage motivation to breastfeed could improve duration of breastfeeding.

and protection against obesity and diabetes. Exclusively breastfed babies have been demonstrated to have higher IQ.[2] They are also less prone to infections, asthma and allergic diseases.[3] Moreover, breastfeeding is beneficial for mothers as research indicates it reduces the risk of breast, uterine and ovarian cancers, postpartum bleeding, and aids postpartum weight loss.[4] Breastfeeding supports short-term and long-lasting health environmental benefits for infants, mothers and the wider population.[5] To recognise the advantages of breastfeeding, political support and financial investment are essential to protect, promote and support breastfeeding.[5]

The WHO recommendations indicate that mothers should provide exclusive breastfeeding to their babies during the first 6 months of life.[6] However, the UK has one of the lowest breastfeeding rates in the

world. 80% of babies are breastfed at birth, but only 1% are exclusively breastfed for 6 months in the UK.[7] These breastfeeding rates are lower among women in areas of higher deprivation and health inequalities.[8] Breastfeeding rates in Wales have remained consistent despite investment in services and UNICEF baby-friendly initiatives.[7] This is a multifaceted issue relating to population health, initiation and maintenance of breastfeeding. It is important to note that breastfeeding rates are influenced by various socioeconomic factors, religion, education and support services available.[8]

Factors that influence women's breastfeeding decisions can include breastfeeding intention, self-efficacy and social support.[9] Exclusive breastfeeding intention describes a mother's intention to provide only breast milk for her baby. Self-efficacy refers to individuals' belief in their ability to complete the necessary behaviours to achieve specific outcomes. Breastfeeding promotion strategies conducted by midwives often include social support; however, there is limited research surrounding intention to breastfeed and self-efficacy and whether this impacts future breastfeeding behaviour.

A systematic review relating to intention to breastfeed has been conducted.[10] Characteristics associated with intention to breastfeed included first-time pregnancy, higher education level, older maternal age, prior breastfeeding experience, non-smoker and living with a partner.[10] Further research has explored prenatal intention to breastfeed as a predictor of exclusive breastfeeding.[11] Intention to breastfeed was a strong predictor of breastfeeding behaviour, with a significant association with exclusive breastfeeding when leaving the hospital after birth.[11] However, this does not indicate whether exclusive breastfeeding continued up to 6 months.

In 2020, the COVID-19 pandemic and concerns about transmission of the disease contributed to higher rates of breastfeeding interruption.[12] Research found that the risk of transmission from infected mothers to their offspring is extremely low.[12] Available data showed that the lack of support for lactating mothers during the pandemic contributed to breastfeeding cessation worldwide.[12] Few strategies were proposed to overcome this issue. Research found that a third of women who planned to breastfeed reported that they did not receive help with positioning the infant.[13] Additionally, one in four women perceived that they did not get enough support with feeding in the hospital.[13] This could have been due to the increased burden on healthcare systems and pressure placed on healthcare professionals to discharge mothers sooner to minimise infection risks, resulting in fewer opportunities to support mothers with infant feeding.

The protection, promotion and support of breastfeeding is a priority for public health. Therefore, encouraging optimal breastfeeding practices including breastfeeding initiation within 1 hour after birth, exclusive breastfeeding for 6 months and continuation of breastfeeding for at least 2 years once nutritious and safe complementary foods are introduced at 6 months is

crucial for meeting the Sustainable Development Goals by 2030.[14]

This study aimed to examine the impact the pandemic had on breastfeeding uptake and duration and whether intention to breastfeed is associated with longer duration of exclusive breastfeeding. Understanding what factors impact exclusive breastfeeding to the recommended 6 months will help develop more targeted interventions to promote and support breastfeeding.

## METHODS
### Study design and setting

A cohort study using routinely collected anonymised population-level linked data from the Secure Anonymised Information Linkage (SAIL) databank.[15 16] Data sources include the Maternal Indicators (MIDS) and the National Community Child Health (NCCH) births and breastfeeding datasets. All women in Wales in the NCCH breastfeeding and MIDS data were deemed eligible for analysis; any stillbirths were excluded. Breastfeeding rates were compared from 2018 to 2021.

### Data sources and linkage

Analysis was undertaken using anonymised population-scale, individual-level linked routinely collected data available in SAIL.[15 16] This anonymously links a wide range of person-based data using a unique, encrypted personal identifier. The linkage in this study includes the MIDS and NCCH births and breastfeeding data. MIDS contains data relating to pregnant women at their initial antenatal assessments and mother and baby (or babies) data including intention to breastfeed at birth for all births. The NCCH comprises information pertaining to birth registration and monitoring of child health examinations. These records were linked for all women known to be pregnant in Wales between 2018 and 2021.

Data in the breastfeeding dataset included the outcome method. Outcomes are categorised in SAIL into exclusively breastfeeding, combined—predominantly breastfeeding, combined—partial breastfeeding and artificial milk. For the purposes of the analyses in this study, the latter three categories were combined as 'other feeds', that is, non-exclusive breastfeeding. The collection times were categorised into first feed, 10 days, 6 weeks and 6 months. Intention to breastfeed was recorded as yes or no; missing breastfeeding outcome and missing MIDS intention were categorised as unknown. Regression analysis was performed on the data. In order to examine how consistent the intention to breastfeed was, we compared the responses from MIDS with responses on the Born in Wales survey.[17] This survey asked expectant mothers how they were planning to feed their baby. Intention during pregnancy (Born in Wales) and after the birth (MIDS) was compared. Informed consent was obtained by an information sheet and consent form provided to participants before starting the survey.

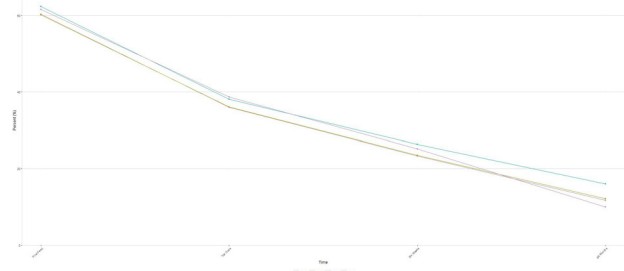

**Figure 1** Study flowchart explaining how the cohort was identified. NCCH, National Community Child Health.

## Patient and public involvement

Midwives from Swansea provided guidance and support during survey design and aided in recruitment by discussing the study at dating scans and routine appointments with expectant mothers.

## Statistical analysis

Descriptive statistics were conducted on rates of breastfeeding per year from 2018 to 2021. Multinomial regression models were used to examine the impact of the explanatory variables intention to breastfeed (assessed on all women attending antenatal visits in Wales), age and ethnicity on breastfeeding rates, reporting ORs with 95% CIs and significance level accepted at a p value of <0.05. The reference groups were those of white ethnicity and those not intending to breastfeed. All data cleaning and analyses with descriptive statistics and regression models were performed in RStudio R V.4.1.3.[18]

## RESULTS

A total of 122 389 babies born to 106 022 mothers were identified from 2018 to 2021. Excluding those babies born outside of Wales (n=3172), stillbirths (n=494) and those having no matching breastfeeding dataset record (n=1692), the cohort comprised n=117 031 babies. Figure 1 describes the participants in the cohort. The mean age of the cohort was 29.4 years. The majority were white (81.9%) and intended to breast feed (58.1%). Table 1 describes the demographics of the cohort.

### Rates of breastfeeding prepandemic, 2020 and 2021

Across the 4 years, between 59.9% and 61.9% of women breastfed their baby for their first feed. This decreased to between 35.1% and 37.3% breastfeeding at 10 days, 25.4%–28.5% at 6 weeks, and those exclusively breastfeeding were lowest at 6 months at 16.6%–20.6% (see figure 2). Between 11.7% and 14% at least partially breastfed at 10 days, 7.5%–8.4% at 6 weeks and 3.4%–4.9% at 6 months. Alternative forms of feeding including a combination of breastmilk and artificial milk, and exclusively artificial milk were mostly adopted after the first feed. The biggest difference was breastfeeding to 6 months between prepandemic (16.6%) and in 2020 (20.6%) (difference: 3.99%, CI 3.35% to 4.62%).

### Impact of intention to breastfeed, age and ethnicity on breastfeeding outcome

The number of women intending to breastfeed did not differ during the pandemic (57.8% of expectant mothers

**Table 1** Demographics of the study cohort

| Category | N | % |
|---|---|---|
| **Intention to breastfeed** | | |
| Yes | 67 973 | 58.1 |
| No | 38 975 | 33.3 |
| NA | 10 083 | 8.6 |
| **Maternal age group** | | |
| 18–24 | 22 719 | 19.4 |
| 25–29 | 34 831 | 29.7 |
| 30–39 | 54 615 | 46.7 |
| 40–50 | 3917 | 3.3 |
| >50 | 10 | 0.0 |
| NA | 1 | 0.0 |
| **Ethnicity** | | |
| White | 95 868 | 81.9 |
| Asian | 3402 | 2.9 |
| Black | 1300 | 1.1 |
| Mixed | 2427 | 2.1 |
| Other | 1169 | 1.0 |
| Unknown | 12 865 | 11.0 |
| Total | 117 031 | 100 |

NA, not available.

intended to breastfeed before the pandemic compared with 58.7% in 2020 (difference: −0.88 (01.58, −0.18%))). Multinomial regression was conducted to examine the variations in breastfeeding rates by intention to breastfeed, adjusted for age, year and ethnicity. Those who intended to breastfeed were 27.6 times more likely to exclusively breastfeed for 6 months compared with those who did not intend to breastfeed (OR 27.6, 95% CI 24.9 to 30.7, p<0.001). Mothers in 2020 were 1.41 times more likely to exclusively breastfeed for 6 months compared with mothers prepandemic (OR 1.41, 95% CI 1.34 to 1.48, p<0.001). Those of black or black British ethnic group were 1.86 times more likely to exclusively breastfeed for 6 months compared with white ethnic group (OR 1.86, 95% CI 1.60 to 2.16, p<0.001; see table 2 and figure 3).

**Figure 2** Exclusive breastfeeding of babies born in Wales between 2018 and 2021

**Table 2** Multinomial regression analysis of factors associated with exclusive breastfeeding outcome at 6 months

| Characteristics | OR (95% CI) |
| --- | --- |
| **Intention to breast feed** | |
| No | Reference |
| Yes | 27.6 (24.9 to 30.7) |
| Unknown | 15.7 (13.9 to 17.6) |
| **Maternal age** | |
| Continuous | 1.06 (1.05 to 1.06) |
| **Ethnic group** | |
| White | Reference |
| Asian | 1.00 (0.90 to 1.11) |
| Black | 1.86 (1.60 to 2.16) |
| Mixed | 1.21 (1.07 to 1.36) |
| Other | 1.27 (1.07 to 1.51) |
| Unknown | 1.24 (1.18 to 1.31) |
| **Year** | |
| 2018 | Reference |
| 2019 | 1.08 (1.02 to 1.13) |
| 2020 | 1.41 (1.34 to 1.48) |
| 2021 | 0.91 (0.86 to 0.96) |

### Born in Wales intention compared with MIDS intention

Of 258 women who completed the Born in Wales survey during pregnancy and had a record in MIDS, 208 (80%) reported an intention to breastfeed. Of these, 88% (183/208) reported the intention to breastfeed again in the MIDS data (after the birth). Twelve per cent (25/208) changed their mind and now intended to not breastfeed or were no longer sure. When a woman reported the intention to not breastfeed on the survey during pregnancy, they changed their mind and reported they would like to breastfeed in less than 9% of cases. This suggests that among this survey population, the initial intention to breastfeed/not breastfeed only changed for about 10% of women.

### DISCUSSION

The findings indicate that approximately 58% of expectant mothers intend to breastfeed and across the 4 years and approximately 60% of mothers in Wales breast-feed for their first feed. However, only 16.6%–20.6% continue to exclusively breastfeed up to the WHO recommended duration of 6 months. The rates of breastfeeding to 6 months were highest during 2020 when the restrictions due to the COVID-19 pandemic were at its height, although intention did not significantly change during 2020. This contrasts with previous research that indicated COVID-19 negatively affected breastfeeding.[12] All predictions were that breastfeeding rates would decline during the pandemic due to reduced access to health professional support and lack of ability for direct social support from family and friends. However, this result suggests that despite the lack of external support options, more mothers were able to invest in exclusive breastfeeding. This might be explained by increased opportunities to remain at home. Having partners at home may have facilitated this. The immunological benefits of breastfeeding may also have contributed to some women pursuing exclusive breastfeeding for longer. However, the rates of exclusive breastfeeding to 6 months are now returning to prepandemic levels.

A systematic review to identify factors influencing exclusive breastfeeding to 6 months has been conducted.[19] Exclusive breastfeeding to 6 months was reportedly affected by maternal working status, breastfeeding knowledge, delivery mode, perception of insufficient human milk, mothers' infant feeding attitude, exclusive breastfeeding self-efficacy and intention.[19] Therefore, breastfeeding to 6 months is associated with working status and breastfeeding knowledge. Supporting women to visualise how they might continue to breastfeed, given their work commitments, for example, where it is feasible to work

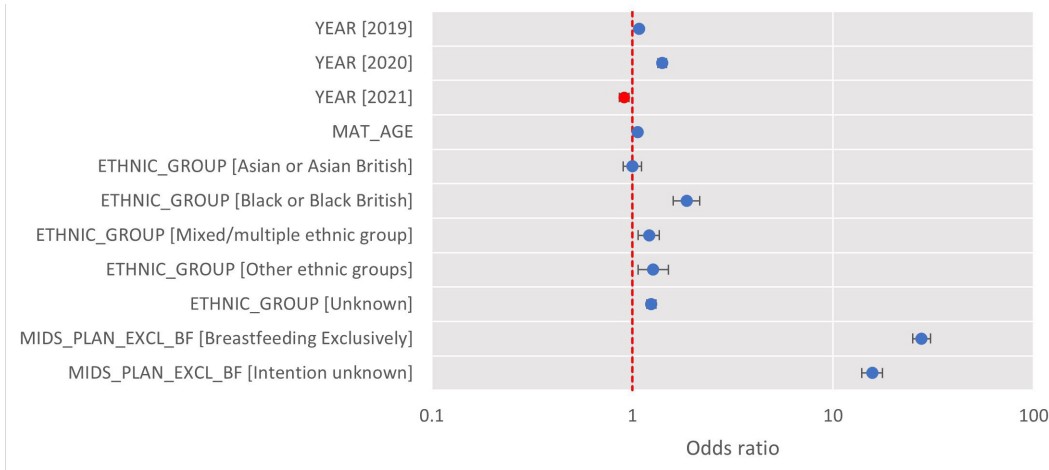

**Figure 3** Multinomial regression for exclusive breastfeeding for 6 months

from home, and improving knowledge early on in pregnancy could help to improve intention to breastfeed for longer. The findings from Born in Wales may also suggest that time with a partner (eg, as seen in COVID-19 lockdown) and so paternal leave may also help with breastfeeding duration. There are many reasons that breastfeeding may not be possible, and so early intervention regarding problem solving and motivation to overcome barriers to breastfeed may facilitate sustained breastfeeding and improve intention to breastfeed. This is suggested by the strong association with intention to breastfeed and a longer duration of breastfeeding, indicating perhaps building motivation and problem solving early is the main facilitator to duration of breastfeeding.

The timing of asking about intention to breastfeed is important to examine. If intention was asked at the start of the pregnancy and this led to a higher uptake of exclusive breastfeeding for 6 months, it would point to the need to influence women's perceptions of breastfeeding even prior to getting pregnant, for example, a social media campaign and advertising. It is rare to see adverts promoting breastfeeding outside of a healthcare setting, but there are adverts for artificial milk. The *Lancet* series on breastfeeding discusses inappropriate marketing tactics of artificial milk companies.[20] The series stresses that sales of artificial milk are directed by well-resourced marketing strategies that represent artificial milk with limited evidence, as recommendations for child health and developmental issues in a manner that undermines breastfeeding.[20] To build an environment for breastfeeding that is not tainted by marketing influences would require stronger political commitment, financial investment and high-level support.

Further investigation would uncover more about the strength of intention and what women understand when asked about their intention to breastfeed. This would clarify whether stating that they intend to breastfeed means they intend to breastfeed immediately after birth, only while in hospital or exclusively for the recommended 6 months.

## Limitations

The study uses healthcare data for pregnant women and new mothers in Wales including maternity and child health data, thus providing a national perspective of breastfeeding rates, enabling the findings to be generalisable. However, it is difficult to know whether the data that were available on breastfeeding are reliable. The quality of data may be influenced by collection methods and how often health visitors were able to visit families to ask questions about methods of feeding and how long they breastfed. Use of expressed milk and donor milk were not explored in the current study, and the datasets employed in this study did not contain information regarding alternative methods. The survey will give a highly optimistic finding as the women who responded to a survey already had higher intention to breastfeed compared with the routine data population, so the findings from this survey will be biased toward breastfeeding.

In conclusion, aspects of the pandemic such as working from home or time with their partner may have improved breastfeeding duration, and so policies and practices that facilitate time with the family may improve duration of breastfeeding. The initiation of breastfeeding was not influenced by the pandemic, so protection from infection did not appear to encourage more women to start breastfeeding. Intention to breastfeed was the strongest predictor of exclusive breastfeeding at 6 months, and therefore, intervention early in the pregnancy or before pregnancy, which facilitates motivation to breastfeed, may improve breastfeeding duration.

**Contributors** HEJ, MJS, MRM and AB: analysis and interpretation of the data. SB: conception and design of the work. HEJ, MJ and NLK: drafting and critical revision of the work for intellectual content. All authors provided substantial contributions this work, contributed to the final approval of the version to be published and agreed to be accountable for all aspects of the work. SB is the guarantor

**Funding** This work is funded by the National Centre for Population Health and Wellbeing Research (grant number AMS103836) and the National Core Studies funded by the Medical Research Council (grant number MC_PC_20030).

**Competing interests** None declared.

**Patient and public involvement** Patients and/or the public were involved in the design, conduct, reporting or dissemination plans of this research. Refer to the Methods section for further details.

**Patient consent for publication** Not applicable.

**Ethics approval** This study, part of Born in Wales and involving human participants, was approved by the Health Research Authority on 16 June 2021 (protocol number RIO 030-20). The participants gave informed consent to participate in the study before taking part. The data held in the SAIL databank in Wales are anonymised. All data contained in SAIL have the permission from the relevant Caldicott guardian or data protection officer.

**Provenance and peer review** Not commissioned; externally peer reviewed.

**Data availability statement** Data are available upon reasonable request. Data can be requested from the Secure Anonymised Information Linkage databank at https://saildatabank.com/

**ORCID iD**
Hope Eleri Jones http://orcid.org/0000-0003-4312-476X

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
