## [Reviewer comments · BMJ Paediatrics Open]

ARTICLE DETAILS

TITLE (PROVISIONAL)	Breastfeeding initiation and duration through the COVID-19 pandemic, a linked population-level routine data study - The Born in Wales cohort 2018-2021
AUTHORS	Jones, Hope Seaborne, Mike Mhereeg, Mohamed James, Michaela Kennedy, Natasha Bandyopadhyay, Amrita Brophy, Sinead

VERSION 1 - REVIEW

REVIEWER	Dr. Helen Louise Ball Durham University, Anthropology
REVIEW RETURNED	19-Feb-2023

GENERAL COMMENTS	This manuscript examines an interesting data set, and some of the results are useful. However the paper's emphasis is prosaic and the main findings could be presented in a more interesting way. The headline finding that 'the biggest predictor of breastfeeding at six months was intention to breastfeed' is redundant. Do we really anticipate that women who expressed no desire to breastfeed would do so? More interesting (perhaps) are those women who expressed a desire to breastfeed and did not do so -- however since the focal question on breastfeeding intent allows for no assessment of strength of intention, or duration of breastfeeding intended very little can be done here (and I strongly encourage changing this question to be more useful for future analyses). The authors also conclude that 'During the pandemic those who intended to breastfeed were more likely to breastfeed for six months compared to before and after the pandemic. Therefore, targeted interventions to encourage intention to breastfeed could improve breastfeeding rates and duration.' This conclusion ignores what it was about the pandemic that facilitated women in meeting their breastfeeding intentions. This could be improved. Comments following the manuscript's chronology: 1. Sentence ending middle of line 6 needs a reference.2. Statement about mathematical abilities in line 8 is unnecessary and should be deleted.3. Sentence ending line 10 needs a reference.4. Line 14 includes an (author, date) style citation that needs converting. This reference does not appear in the bibliography.5. Line 35, suggest changing 'minimal' to limited.6. The research that DOES exist around intention to breastfeed should be acknowledged here.7. Line 61 - replace 'If' with 'whether'8. Line 62 - what does 'higher levels' of breastfeeding mean? Please use a more precise phrase.
--

9. Line 62 - replace 'If' with 'whether'
10. Line 64 - clarify that the recommendation is for exclusive breastfeeding to 6 months, not just any breastfeeding.
11. Line 77 - the assumption that 'as the mother's outcome of breastfeeding and intention to feed at each birth is independent of the decisions at another birth' is incorrect. Multiple studies indicate prior breastfeeding experience predicts future breastfeeding duration -- see for example:
<https://bmjopen.bmj.com/content/3/8/e003274> and
<https://internationalbreastfeedingjournal.biomedcentral.com/articles/10.1186/s13006-021-00364-6>. The authors should reconsider whether inclusion of multiparous women in their analysis is appropriate.
12. Line 94 -- the yes/no nature of this question is a major limitation -- some indication of strength of intent, and intended duration would be very useful additions to this data for future work.
13. Line 117 -- is the section in this sentence stating 'and that there was no other outcome to be coded' really necessary? Seems redundant.
14. Line 135 - delete 'of'. The sample comprised 117,000 babies.
15. Line 137 - inappropriate sentence break.
16. Line 147 - the data contained in this section could be usefully illustrated graphically.
17. Line 154 - a 4% increase in exclusive breastfeeding at 6 months is an important finding. This should be the headline of your paper. All predictions were that breastfeeding rates would decline during the pandemic due to reduced access to health professional support, and lack of ability for direct social support from family and friends. However this result suggests that despite the lack of external support options, the ability to remain at home meant more mothers were able to invest in exclusive breastfeeding. Having partners at home may have facilitated this. The immunological benefits of breastfeeding may also have contributed to some women pursuing exclusive breastfeeding for longer.
18. Line 165 - to breastfeed to six months or to exclusively breastfeed to six months? Please also clarify for the remainder of this paragraph.
19. Line 173 - please clarify whether Table 2 shows outcomes for exclusive breastfeeding to 6 months or any breastfeeding to 6 months.
20. Line 183 - amend 'among a survey population' to 'among this survey population'
21. Line 184 - amend 'changes' to 'changed'
22. Line 197 - amend 'breastfeed' to 'exclusive breastfeeding'
23. Line 206 - working status differs from employment status. Here you mean working status. What differed during the pandemic was that employed women were more frequently not working (furloughed) or were working from home for longer than maternity leave may have otherwise allowed
24. Line 205 - exclusive??
25. Line 206 - as above re employment vs. working
26. Line 210 - this section could usefully discuss leaning more about strength of intention and what women understand by the question 'do you intend to breastfeed?' and whether their 'Yes' response means 'once immediately after birth', 'yes while in hospital' etc.
27. Line 215 - make reference here to the recent Lancet series on breastfeeding discussing the inappropriate marketing tactics of formula companies.
28. Line 232 - to me this is not the most important finding or conclusion from this paper. Also as the small survey study adds nothing to the conclusions presented I am left wondering what it adds to this article and why it is included. Please justify this or delete reference to it.
29. Line 232 completely overlooks women's ABILITY to exclusively breastfeed to 6 months as suggested by the pandemic data -- make more of this in the conclusions!

REVIEWER	Dr. Peter Flom Peter Flom Consulting
REVIEW RETURNED	23-Feb-2023

GENERAL COMMENTS	I confine my remarks to statistical and methodological aspects of this paper. The general approach is OK, but I have quite a few issues to resolve before I can recommend publication. General: The authors should avoid causal language (e.g. "impacted") since this is an observational study. The conclusion in the abstract seems sensible and is *indicated* by the results, but it doesn't directly follow from them. This is because causation has not been established. To take an extreme example, shark attacks increase when ice cream is eaten. But discouraging the eating of ice cream will have no effect on shark attacks. Now, that's ridiculous and the authors' example is not. Nevertheless, it could be, for example, that encouraging intention to breastfeed makes women resentful and makes them *less* likely to breastfeed. More generally, women who are encouraged to breastfeed may not be the same as women who intend to breastfeed without encouragement. Also, speaking from personal experience of my ex-wife, many women who don't breastfeed have good reasons for not doing so (in her case, she was taking medications at the time of birth that would have had very bad effects on the baby). Encouraging these women to breastfeed would have no effect. Line 96-97 (and thank you for providing these line numbers!) I think there is a typo here, but I am not sure what it is. Right now, I can't figure out what it means. Maybe it should be "... to 258 questions"? Line 114 Age should not be categorized or grouped. This increases type 1 and type 2 error. Leave age in years and use splines to evaluate nonlinearity. See my blog post https://medium.com/@peterflom/what-happens-when-we-categorize-an-independent-variable-in-regression-77d4c5862b6c Line 116 Since you have the entire population, some statisticians (including me) would argue that p values and CIs make no sense. p values and CIs are about inference from a sample to a population. You have the population. Other statisticians argue that you can posit some sort of "super population". Although I don't think this makes much sense in most cases, I won't insist. But the authors should deal with this question. Line 136 Why was this missing? With this much missing data, simply deleting cases could be problematic. It would be much better to use multiple imputation. An alternative is to include NA as a third category. This might be the only choice available, if a lot of data is missing. Table 1 Please include mean age (or median). It seems like the study population was a bit older than the full cohort. It also seems like they were less likely to be non-white. This means that simply dropping the missing cases is not a good idea.
---

	Not exactly a statistical point, but it seems to me that lines 148 to 152 show that a key factor in increasing breastfeeding is inducing women who do so at first feed to continue doing so. I'm not sure how this could be related to intention, but a later study could ask more detailed intention questions, including questions about first feeding and later feeding and 6 month feeding. Line 163 No R square measure for logistic regression is equivalent to "% of variation". See e.g https://statisticalhorizons.com/r2logistic/ which is an article by Paul Allison, one of the leading experts on logistic reg. Figure 2 needs a label for the x-axis (probably "odds ratio")
--	---

VERSION 1 – AUTHOR RESPONSE

Reviewer 1

This manuscript examines an interesting data set, and some of the results are useful. However the paper's emphasis is prosaic and the main findings could be presented in a more interesting way. The headline finding that 'the biggest predictor of breastfeeding at six months was intention to breastfeed' is redundant. Do we really anticipate that women who expressed no desire to breastfeed would do so? More interesting (perhaps) are those women who expressed a desire to breastfeed and did not do so -- however since the focal question on breastfeeding intent allows for no assessment of strength of intention, or duration of breastfeeding intended very little can be done here (and I strongly encourage changing this question to be more useful for future analyses).

The authors also conclude that 'During the pandemic those who intended to breastfeed were more likely to breastfeed for six months compared to before and after the pandemic. Therefore, targeted interventions to encourage intention to breastfeed could improve breastfeeding rates and duration.' This conclusion ignores what it was about the pandemic that facilitated women in meeting their breastfeeding intentions. This could be improved.

Many thanks for this feedback. We have redrafted the conclusions and title of the paper to take into account these recommendations. See below.

Title

Breastfeeding initiation and duration through the COVID-19 pandemic, a linked population-level routine data study - The Born in Wales cohort 2018-2021

Abstract:

This study aimed to examine the impact the pandemic had on breastfeeding uptake and duration and if intention to breastfeed is associated with longer duration of exclusive breastfeeding.

Conclusion: Women were more likely to exclusively breastfeed for six months during the pandemic compared to before or after the pandemic. Arguably, interventions which enable families to spend more time with their baby such as maternal and paternal leave may help improve breastfeeding duration. The biggest predictor of breastfeeding at six months was intention to breastfeed. Therefore, targeted interventions during pregnancy to encourage motivation to breastfeed could improve duration of breastfeeding.

Page 15: In conclusion, aspects of the pandemic such as working from home or time with a partner may have improved breastfeeding duration and so policies and practices that facilitate time with the

family may improve duration of breastfeeding. The initiation of breastfeeding was not influenced by the pandemic so protection from infection did not appear to encourage more women to start breastfeeding. Intention to breastfeed was the strongest predictor of exclusive breastfeeding at six months and therefore intervention early in the pregnancy or before pregnancy which facilitates motivation to breastfeed may improve breastfeeding duration.

Comments following the manuscript's chronology:

1. Sentence ending middle of line 6 needs a reference.

Reference has been added

2. Statement about mathematical abilities in line 8 is unnecessary and should be deleted.

This has been deleted

3. Sentence ending line 10 needs a reference.

Reference has been added

4. Line 14 includes an (author, date) style citation that needs converting. This reference does not appear in the bibliography.

This has been converted

5. Line 35, suggest changing 'minimal' to limited.

This has been changed

6. The research that DOES exist around intention to breastfeed should be acknowledged here.

Existing research around intention to breastfeed has been added here

7. Line 61 - replace 'If' with 'whether'

The wording has been changed here

8. Line 62 - what does 'higher levels' of breastfeeding mean? Please use a more precise phrase.

This has been amended to 'the impact the pandemic had on breastfeeding uptake and duration and if intention to breastfeed is associated with longer duration of exclusive breastfeeding'.

9. Line 62 - replace 'If' with 'whether'

The wording has been changed here

10. Line 64 - clarify that the recommendation is for exclusive breastfeeding to 6 months, not just any breastfeeding.

This has been clarified

11. Line 77 - the assumption that 'as the mother's outcome of breastfeeding and intention to feed at each birth is independent of the decisions at another birth' is incorrect. Multiple studies indicate prior breastfeeding experience predicts future breastfeeding duration -- see for example:

<https://eur03.safelinks.protection.outlook.com/?url=https%3A%2F%2Fbmjopen.bmj.com%2Fcontent%2F3%2F8%2F003274&data=05%7C01%7CM.L.James%40Swansea.ac.uk%7Ca10c9cf8651b4bc869a208db170f555c%7Cbbcab52e9fbe43d6a2f39f66c43df268%7C0%7C0%7C638129126842895723%7CUnknown%7CTWFpbGZsb3d8eyJWljojMC4wLjAwMDAiLCJQIjoiV2luMzliLCJBTiI6Ikl1haWwiLCJ>

XVCI6Mn0%3D%7C3000%7C%7C%7C&sdata=NLWgfaltxUqIRUsN7YEI3zwo5zs7poO%2FFWR208hS1U%3D&reserved=0 and

<https://eur03.safelinks.protection.outlook.com/?url=https%3A%2F%2Finternationalbreastfeedingjournal.biomedcentral.com%2Farticles%2F10.1186%2Fs13006-021-00364-6&data=05%7C01%7CM.L.James%40Swansea.ac.uk%7Ca10c9cf8651b4bc869a208db170f555c%7Cbbcab52e9fbe43d6a2f39f66c43df268%7C0%7C0%7C638129126842895723%7CUnknown%7CTWFpbGZsb3d8eyJWljiMC4wLjAwMDAiLCJQIjoiV2luMzliLCJBTiI6IjEhaWwiLCJXVCI6Mn0%3D%7C3000%7C%7C%7C&sdata=KkGs8kw9g7KLFTZHWd3gM47IN7sRT8v1Xqv55YFBj3s%3D&reserved=0>. The authors should reconsider whether inclusion of multiparous women in their analysis is appropriate.

Thank you for this comment. We have removed this assumption

12. Line 94 -- the yes/no nature of this question is a major limitation -- some indication of strength of intent, and intended duration would be very useful additions to this data for future work.

We have used the question asked routinely by the midwives to all women in Wales, so this was not a question we could amend or change. We would agree that intention duration would be useful and will pass this information on to the early years team in Public Health Wales.

13. Line 117 -- is the section in this sentence stating 'and that there was no other outcome to be coded' really necessary? Seems redundant.

This has been removed

14. Line 135 - delete 'of'. The sample comprised 117,000 babies.

This has been deleted

15. Line 137 - inappropriate sentence break.

This has been amended

16. Line 147 - the data contained in this section could be usefully illustrated graphically.

A graph has been added to illustrate the exclusive breastfeeding rates over the four years (this is now Figure 2)

17. Line 154 - a 4% increase in exclusive breastfeeding at 6 months is an important finding. This should be the headline of your paper. All predictions were that breast-feeding rates would decline during the pandemic due to reduced access to health professional support, and lack of ability for direct social support from family and friends. However this result suggests that despite the lack of external support options, the ability to remain at home meant more mothers were able to invest in exclusive breastfeeding. Having partners at home may have facilitated this. The immunological benefits of breastfeeding may also have contributed to some women pursuing exclusive breastfeeding for longer.

We have amended the discussion to reflect this advice

Page 13

All predictions were that breastfeeding rates would decline during the pandemic due to reduced access to health professional support, and lack of ability for direct social support from family and friends. However, this result suggests that despite the lack of external support options, more mothers were able to invest in exclusive breastfeeding. This might be explained by increased opportunities to remain at home. Having partners at home may have facilitated this. The immunological benefits of

breastfeeding may also have contributed to some women pursuing exclusive breastfeeding for longer. However, the rates of exclusive breastfeeding to six months are now returning to pre pandemic levels.

18. Line 165 - to breastfeed to six months or to exclusively breastfeed to six months? Please also clarify for the remainder of this paragraph.

This has been clarified as exclusively breastfeed to six months through this paragraph

19. Line 173 - please clarify whether Table 2 shows outcomes for exclusive breastfeeding to 6 months or any breastfeeding to 6 months.

This has been clarified that Table 2 shows outcomes for exclusive breastfeeding to 6 months

20. Line 183 - amend 'among a survey population' to 'among this survey population'

This has been amended

21. Line 184 - amend 'changes' to 'changed'

This has been amended

22. Line 197 - amend 'breastfeed' to 'exclusive breastfeeding'

This has been amended

23. Line 206 - working status differs from employment status. Here you mean working status. What differed during the pandemic was that employed women were more frequently not working (furloughed) or were working from home for longer than maternity leave may have otherwise allowed

This has been amended to working status

24. Line 205 - exclusive??

The word exclusive has been added here

25. Line 206 - as above re employment vs. working

This has been amended to working status

26. Line 210 - this section could usefully discuss learning more about strength of intention and what women understand by the question 'do you intend to breastfeed?' and whether their 'Yes' response means 'once immediately after birth', 'yes while in hospital' etc.

Discussion about the strength of intention and interpretation of the question 'Do you intend to breastfeed?' has been added

27. Line 215 - make reference here to the recent Lancet series on breastfeeding discussing the inappropriate marketing tactics of formula companies.

Detail from the Lancet series on breastfeeding has been added here

28. Line 232 - to me this is not the most important finding or conclusion from this paper. Also as the small survey study adds nothing to the conclusions presented I am left wondering what it adds to this article and why it is included. Please justify this or delete reference to it.

We have added the following to the methods:

In order to examine how consistent the intention to breastfeed was we compared the responses from MIDS with responses on the Born in Wales survey

29. Line 232 completely overlooks women's ABILITY to exclusively breastfeed to 6 months as suggested by the pandemic data -- make more of this in the conclusions!

The conclusions have been amended to reflect this. See below

In conclusion, aspects of the pandemic such as working from home or time with their partner may have improved breastfeeding duration and so policies and practices that facilitate time with the family may improve duration of breastfeeding. The initiation of breastfeeding was not influenced by the pandemic so protection from infection did not appear to encourage more women to start breastfeeding.

Reviewer 2

I confine my remarks to statistical and methodological aspects of this paper. The general approach is OK, but I have quite a few issues to resolve before I can recommend publication.

General: The authors should avoid causal language (e.g. "impacted") since this is an observational study.

Thank you for this comment. We have amended any causal language through this manuscript to address this

The conclusion in the abstract seems sensible and is *indicated* by the results, but it doesn't directly follow from them. This is because causation has not been established. To take an extreme example, shark attacks increase when ice cream is eaten. But discouraging the eating of ice cream will have no effect on shark attacks. Now, that's ridiculous and the authors' example is not. Nevertheless, it could be, for example, that encouraging intention to breastfeed makes women resentful and makes them *less* likely to breastfeed. More generally, women who are encouraged to breastfeed may not be the same as women who intend to breastfeed without encouragement.

Also, speaking from personal experience of my ex-wife, many women who don't breastfeed have good reasons for not doing so (in her case, she was taking medications at the time of birth that would have had very bad effects on the baby). Encouraging these women to breastfeed would have no effect.

We have amended our conclusions in line with response to reviewer 1. In addition we have added the following to the discussion:

There are many reasons that breastfeeding may not be possible and so early intervention regarding problem solving and motivation to overcome barriers to breastfeed may facilitate sustained breastfeeding and improve intention to breastfeed. This is suggested by the strong association with intention to breastfeed and a longer duration of breastfeeding, suggesting perhaps building motivation and problem solving early is the main facilitator to duration of breastfeeding.

Line 96-97 (and thank you for providing these line numbers!) I think there is a typo here, but I am not sure what it is. Right now, I can't figure out what it means. Maybe it should be ".... to 258 questions"?

To clarify, this is 258 responses to the Born in Wales survey. The wording has been amended

Line 114 Age should not be categorized or grouped. This increases type 1 and type 2 error. Leave age in years and use splines to evaluate nonlinearity. See my blog post <https://eur03.safelinks.protection.outlook.com/?url=https%3A%2F%2Fmedium.com%2F%40peterflom%2Fwhat-happens-when-we-categorize-an-independent-variable-in-regression->

77d4c5862b6c&data=05%7C01%7CM.L.James%40Swansea.ac.uk%7Ca10c9cf8651b4bc869a208db170f555c%7Cbbcab52e9fbe43d6a2f39f66c43df268%7C0%7C0%7C638129126842895723%7CUnknown%7CTWFpbGZsb3d8eyJWljojMC4wLjAwMDAiLCJQIjoiV2luMzliLCJBTiI6IjEhaWwiLCJXVCi6Mn0%3D%7C3000%7C%7C%7C&sdata=anjkkQSSB5NO0%2BoMkcLEvaMWwdRpF9WAksCV2bA4%2FIE%3D&reserved=0

Maternal age has been changed to a continuous variable for regression analysis

Line 116 Since you have the entire population, some statisticians (including me) would argue that p values and CIs make no sense. p values and CIs are about inference from a sample to a population. You have the population. Other statisticians argue that you can posit some sort of "super population". Although I don't think this makes much sense in most cases, I won't insist. But the authors should deal with this question.

We would agree this was the entire population of Wales and so was not a sample in terms of Wales. However, in order to give estimates for other countries like England, Scotland or N.Ireland we feel it is a sample for the UK, so would feel that the CI's make sense in terms of inference from a Wales sample to a UK population.

Line 136 Why was this missing? With this much missing data, simply deleting cases could be problematic. It would be much better to use multiple imputation.

We do not have details on why this data is missing from the database. We have changed this to have the NA's as a category rather than removing them

Table 1 Please include mean age (or median).

The mean age has been added

It seems like the study population was a bit older than the full cohort. It also seems like they were less likely to be non-white. This means that simply dropping the missing cases is not a good idea.

Not exactly a statistical point, but it seems to me that lines 148 to 152 show that a key factor in increasing breastfeeding is inducing women who do so at first feed to continue doing so. I'm not sure how this could be related to intention, but a later study could ask more detailed intention questions, including questions about first feeding and later feeding and 6 month feeding.

Thank you for this comment. We have added detail to the discussion about further investigation into strength of intention and continuing to feed for a longer duration

Line 163 No R square measure for logistic regression is equivalent to "% of variation". See e.g. <https://eur03.safelinks.protection.outlook.com/?url=https%3A%2F%2Fstatisticalhorizons.com%2F2logistic%2F&data=05%7C01%7CM.L.James%40Swansea.ac.uk%7Ca10c9cf8651b4bc869a208db170f555c%7Cbbcab52e9fbe43d6a2f39f66c43df268%7C0%7C0%7C638129126842895723%7CUnknown%7CTWFpbGZsb3d8eyJWljojMC4wLjAwMDAiLCJQIjoiV2luMzliLCJBTiI6IjEhaWwiLCJXVCi6Mn0%3D%7C3000%7C%7C%7C&sdata=%2B4Y3b6qbxQaVrrfxpod6hXKAASIHjgaRrAy2UraECDY%3D&reserved=0> which is an article by Paul Allison, one of the leading experts on logistic reg.

Thank you for this comment, this statement has been removed

Figure 2 needs a label for the x-axis (probably "odds ratio")

This has been added (This is now Figure 3)

VERSION 2 – REVIEW

REVIEWER	Dr. Peter Flom Peter Flom Consulting
REVIEW RETURNED	17-Mar-2023

GENERAL COMMENTS	The authors have addressed my concerns and I now recommend publication
--

REVIEWER	Dr. Helen Louise Ball Durham University, Anthropology
REVIEW RETURNED	30-Mar-2023

GENERAL COMMENTS	Thank you for responding to all of my comments and suggestions and for modifying the manuscript accordingly. I think the changes to the title, abstract and conclusion are positive. But I am still struggling to understand why the key highlights still emphasise an outcome that basically says that if someone intends to do something they are much more likely to be doing it 6 months later than someone who had no intention of ever starting to do it. Is this not self-evident? Your data about what happened during the pandemic seems far more interesting in terms of understanding what facilitates breastfeeding to 6 months, than your data confirming the fact that when people decide not to do something they don't do it! Please review your manuscript for the use of 'if' vs 'whether'. I suggested changes in the previous draft, but now new instances of 'if' have been added in places that to me are incorrect. I would suggest the phrase 'data wrangling' is informal terminology and should not be used in academic writing.
--

VERSION 2 – AUTHOR RESPONSE

Reviewer 2

Comments to the Author

Thank you for responding to all of my comments and suggestions and for modifying the manuscript accordingly. I think the changes to the title, abstract and conclusion are positive. But I am still struggling to understand why the key highlights still emphasise an outcome that basically says that if someone intends to do something they are much more likely to be doing it 6 months later than someone who had no intention of ever starting to do it. Is this not self-evident? Your data about what happened during the pandemic seems far more interesting in terms of understanding what facilitates breastfeeding to 6 months, than your data confirming the fact that when people decide not to do something they don't do it!

The key messages have been amended to include 'Breastfeeding rates at 6 months were significantly higher during the COVID-19 pandemic. Black mothers were significantly more likely to exclusively breastfeed for 6 months.'

This research was requested by the Midwives we work with, they ask women about intention to breastfeed but to date it has not been examined whether the intention translates to higher actual

breastfeeding outcomes or if intention/non-intention is highly changeable. We would agree looking in retrospect it looks self-evident, but before the study it was not self-evident to everyone. As the midwives asked us to explore this, we want to keep these findings in the manuscript.

Please review your manuscript for the use of 'if' vs 'whether'. I suggested changes in the previous draft, but now new instances of 'if' have been added in places that to me are incorrect.

These instances have been amended

I would suggest the phrase 'data wrangling' is informal terminology and should not be used in academic writing.

This wording has been changed to 'data cleaning'